# Effects, Doses, and Applicability of Gestrinone in Estrogen-Dependent Conditions and Post-Menopausal Women

**DOI:** 10.3390/ph17091248

**Published:** 2024-09-22

**Authors:** Guilherme Renke, Mariana Antunes, Renato Sakata, Francisco Tostes

**Affiliations:** Nutrindo Ideais Performance and Nutrition Research Center, Rio de Janeiro 22411-040, Brazil; dramarianantunes@gmail.com (M.A.); renatosakata@gmail.com (R.S.); ftostes@msn.com (F.T.)

**Keywords:** gestrinone, endometriosis, hormone replacement therapy, implant, menopause

## Abstract

Gestrinone (R-2323), or ethylnorgestrienone, is a synthetic steroid of the 19-nortestosterone group more commonly used as an oral, intravaginal, or subcutaneous implant for the treatment of endometriosis, contraception, and estrogen-dependent conditions such as hypermenorrhea, premenstrual dysphoria, and intense menstrual cramps. This review aims to reevaluate the routes, doses, and applicability proposed for using gestrinone, including its use in new conditions such as menopause, lipedema, and sarcopenia. Here, we present the possible application of gestrinone as a long-acting therapeutic possibility through hormonal implants and the benefits and potential risks. Available evidence on the safety of doses and routes is limited. Gestrinone appears to be effective compared to other progestins and may have some advantages in the treatment of estrogen-dependent pathologies. Future research must evaluate gestrinone’s long-term safety and potential therapeutic indications.

## 1. Introduction

Gestrinone is a 19-norsteroid with the characteristics of having an affinity for progesterone receptors and acting as an antagonist, becoming an anti-progesterone molecule. Furthermore, having anti-androgenic and anti-aldosterone characteristics, this molecule has an affinity for aldosterone and androgen receptors. On the other hand, its most striking and accentuated characteristic is its potent anti-estrogenic action, which is why it is widely studied and used to treat estrogen-dependent pathologies [1]. The molecule also acts on the nuclear factor kappa-light-chain-enhancer of activated B cell (NF-Kβ) receptors and reduces aromatase’s action due to its anti-estrogenic potential. It has anti-inflammatory action on tissues, mainly in the uterus, making it widely applicable for the treatment of adenomyosis and endometriosis [1].

Gestrinone is a progestin that has an antagonistic action on glucocorticoid receptors and gonadotropin-leading hormone (GnRH) receptors and is antagonistic to steroid hormone-binding globulin (SHBG). Like other progestins, the molecule has an antagonist/agonist action against estrogen receptors. However, its potency as an anti-estrogen agent is increased by the action of the enzymes 17β-hydroxysteroid dehydrogenase and aromatase.

The characteristics of the R-2323 molecule confer it with some interesting metabolic effects and clinical applicability concerning other progestins (Table 1). These include menstrual block, the inhibition of ovulation, endometrial atrophy, and the atrophy of breast tissue; it has a lipolytic effect by blocking the adipogenic cascade through inhibiting aromatase in the adipocyte [2,3,4].

Previous studies have also analyzed the anti-cancer effects of gestrinone in women with breast, endometrial, ovarian, hepatic, and cervical cancer [5,6,7]. The impact of gestrinone was observed to induce apoptosis of HeLa cells, with tumor-suppressive action through the c-Jun N-terminal kinase (JNK-p21) pathway; it was also shown to reduce the expression of m-RNA and oxidative stress in the endoplasmic reticulum [5].

## 2. Material and Methods

The presented scientific literature was reviewed and studies were downloaded from PubMed (https://pubmed.ncbi.nlm.nih.gov/ (accessed on 28 July 2024)), ScienceDirect (https://sciencedirect.com/ (accessed on 28 July 2024)), and SciELO (https://scielo.com.br/ (accessed on 28 July 2024)) databases. Combinations of several search terms—such as “gestrinone”, “ethylnorgestrienone”, “R-2323”, “endometriosis”, “contraception”, “lipedema”, “hypermenorrhea”, “premenstrual dysphoria”, “sarcopenia” and “menopause”—were applied. After the search, the studies were classified according to the health-specific parameters of the text. Our selected primary outcomes were changes in estrogen-dependent conditions in pre- and post-menopausal women. Secondary outcomes included doses, routes, hormonal parameters, and adverse events.

## 3. Results and Discussion

Preclinical data from studies evaluated the effect of gestrinone on the viability of breast cancer (MDA-MB-231) cells and liver cancer (Huh7) cells. These two types of cell lines were exposed to different concentrations of gestrinone to understand the possible mechanism of action of gestrinone. In MDA cells, a reduction in cell viability was observed in a concentration- and time-dependent manner. However, this decrease appeared to be more significant than that in Huh7, suggesting that this cell type may be more sensitive to treatment with gestrinone [6]. The breast cells used in this trial, MD-MB-231, were obtained from the subtype of breast cancer known as triple-negative. This term is used because these tumor cells do not have receptors for estrogen or progesterone, in addition to not producing the HER2 protein, which makes it particularly aggressive, precisely due to the absence of specific therapeutic targets and limited treatment options [7].

In Huh7 cells, there was a reduction in cell viability in a concentration-dependent manner in the 48 h and time-dependent assay, except in the condition of exposure to a concentration of 1 µmol/L of gestrinone. In MDA-MB3 mammary cells, cell viability was reduced in a concentration- and time-dependent manner. Cell viability over 24 and 48 h was determined by a methylthiazolyl tetrazolium (MTT) assay; the cells were incubated in the presence of increasing concentrations of gestrinone at 1, 5, 10, 50, and 100 µmol/L [6].

Concerning breast cancer, estrogen is responsible for ductal growth of the breast, while progesterone leads to alveolar differentiation. In turn, testosterone replacement in women of both childbearing age and menopause was shown to have a protective effect on the breast in long-term retrospective studies [8]. Some authors even propose its use as a therapeutic option for invasive breast cancer, either alone, or associated with an aromatase inhibitor (anastrozole) to reduce the conversion of testosterone into estrogen [9]. More specifically, evidence suggests that the testosterone–androgen receptor complex is antiproliferative and opposes estrogen’s stimulatory effects, increasing the apoptosis rate of breast cancer cell lines [10,11]. It is essential to highlight that these effects of testosterone were not observed in other routes of administration, such as the transdermal route, suggesting that implant therapy appears to promote a more constant release and more stable plasma levels of the hormone [12].

Therefore, the reduction in the viability of this cell line after exposure to gestrinone suggests that this effect of the hormone does not depend on its anti-estrogenic action, as these cells are negative for estrogen receptors (Figure 1). It may be related precisely to its agonist effect on the androgen receptor in the breast, similar to that which occurs in endometriotic lesions. However, the protective effect of gestrinone on the breast is not exclusively due to its action on the androgen receptor; another indication for the use of gestrinone is fibrocystic breast disease, the most common disorder of the human female breast. It is estimated that 30% of women between 35 and 50 years old develop some form of the disease. Its importance cannot be underestimated, as malignant breast tumors are three times more common in patients affected by this condition [4]. Gestrinone is indicated in this breast pathology as it occupies the estrogen and progesterone receptors, reducing the inflammatory process and breast volume, the size and quantity of cysts and nodules, and mastalgia [4].

The effect of different progestins, including gestrinone, on breast diseases, especially breast cancer, is still controversial. The type of progestin, as well as its dose, duration, and use, with or without estrogen, are determining factors that define whether the effect will be proliferative or apoptotic in breast tissue [13]. However, gestrinone’s androgenic and antiestrogenic effects, its antiproliferative and anti-inflammatory action in fibrocystic breast disease, and our results in breast cells may indicate its safety and its potential protective effects on the breast.

Other authors’ findings have reinforced gestrinone’s antiproliferative and anticancer effects in inhibiting uterine leiomyoma cells [14] and cervical cancer cells [5] in a colorimetric assay for assessing cell viability trials after exposure to gestrinone concentrations. These findings are similar to those of other authors who used chemotherapeutic, anti-proliferative, and anti-angiogenic agents, providing an interesting perspective regarding gestrinone’s safety and potential therapeutic indications [7].

Regarding endometriosis, which is one of the most common gynecological pathologies, affecting around 6–10% of women of reproductive age (Figure 1), there is a clear benefit from gestrinone. It is known that in this condition, estrogen levels fuel the growth and maintenance of the ectopic endometrium and lead to an increase in endometriotic foci due to steroidogenesis [5]. Endometriosis is known to be one of the causes of female infertility. In addition to causing symptoms such as intense pelvic pain, anxiety, and depression, it also increases the rate of hospitalizations and recurrence surgeries [5,15].

The treatment of endometriosis is complex in many cases because when there is a need for surgical intervention, there is a shortage of professionals qualified to perform the appropriate techniques; the drug efficacy of clinical treatments is still the subject of great discussion in the medical community. It is considered a pelvic pathology of an inflammatory nature with an increased production of pro-inflammatory cytokines and chemokines, culminating in a significant increase in estrogen production that feeds inflammation by feedback through an uncontrolled rise in prostaglandins levels and the potent mitogenic effect of estradiol [5].

In this scenario, the antiestrogenic and anti-inflammatory characteristics of gestrinone efficiently treat endometriotic foci and pelvic pain resulting from the disease. Studies comparing the long-term use of gestrinone versus GNRH analogs show the superiority of gestrinone as part of the therapeutic arsenal when treating endometriosis [15]. Furthermore, the immunological effects of gestrinone in regulating tumor necrosis factor-alpha (TNF-α), in inhibiting NF-Kβ, and in reducing monocyte phagocytosis and IL-10 and IL-4 make it a molecule with antiproliferative and anti-inflammatory potential, with applicability not only in the clinical treatment of endometriosis but also in other pathologies that occur with inflammatory disorders [16]. In agreement with this, the European Society of Human Reproduction and Embryology (ESHRE) guideline points to gestrinone as a valid therapeutic option for endometriosis [17].

The effect of the regression of the uterine size of women undergoing treatment with gestrinone to reduce leiomyomas makes the molecule of significant attractiveness and importance in our clinical practice for use in the hormonal therapy of climacteric and post-menopause patients who have a history of estrogen-dependent pathologies [18].

Gestrinone, as a molecule from the progestin class, has the potential to play a role in endometrial protection during hormone therapy (HT) in post-menopausal women when used in appropriate doses; it is associated with estrogen replacement due to its ability to promote endometrial atrophy [18]. A characteristic of the molecule is to promote a prolonged period of amenorrhea, which motivated the doctor and scientist Elsimar Coutinho, the developer of gestrinone silastic implant technology in the 1970s, to determine the ideal annual dose of the silicone implant to promote contraception for 12 months with few side effects [1].

A multicenter study compared the effectiveness of an average of 210 mg of gestrinone per study participant, administered via silastic subcutaneous implants, proving that its contraceptive effect is as practical as the levonorgestrel implant. One of the characteristics of gestrinone is a prolonged time of amenorrhea, which makes it a valuable tool for treating chronic anemia resulting from heavy menstrual flows [1,19].

The remarkable effectiveness of the molecule in the treatment of menstrual dysphoric disorder is also observed in patients who require contraception and who are elective for the use of gestrinone. As a progestin, it acts on the central nervous system, playing a role in improving the symptoms of mood disorders [19]. This effect is similar to that of another progestin, nestorone, which has a possible neuroprotective effect [20]. Nestorone has neuroprotective potential due to its effective binding to brain progesterone receptors (PRs) [21]. PRs that perform different regulatory functions are found in the adult human brain, which makes their activation interesting with respect to brain repair [22]. As it is a specific agonist that selectively binds to PRs and does not bind significantly to glucocorticoid receptors and androgen receptors, nestorone expresses a potent binding to PRs, which gives the substance the characteristic of acquiring a high affinity even at low doses, providing the medicine with a potential neuroprotective role. Its oral administration in animals or humans is inactive but it is potentially active when administered subcutaneously or transdermally [21,22]. Thus, due to the similarity between nestorone and gestrinone, future studies evaluating the real benefit of gestrinone in neuroprotection are of interest.

Menstrual dysphoric disorder is one of the estrogen-dependent pathologies for which gestrinone is an exciting and significant therapeutic possibility. However, gestrinone does not directly interfere with the increase in total testosterone levels, nor even with the reduction in total serum testosterone. However, we have observed decreased SHBG levels in patients [23]. The intrinsic androgenic ability of the molecule can cause side effects, such as seborrhea, acne, increased skin oiliness, hair loss, hoarseness, increased libido, and clitoral hypersensitivity, in some women who are more sensitive to the metabolism of androgenic hormones.

The action of gestrinone on estrogen receptors in the breast tissue has been demonstrated to be efficient, in low doses of around 10 mg/week, over a short period of treatment, of around 2–3 months, at regressing benign breast fibroadenomas. Fibrocystic breast disease is a common female disorder, and we should not underestimate it, as we know that malignant tumors are more common in breasts affected by this pathology [4].

The most-reported side effects from the use of gestrinone always correlate with increased skin oiliness, acne and seborrhea, and weight gain. However, we cannot ignore the anabolic effect it provides, which is desired by older women who present with reduced muscle mass and sarcopenia [4].

Another disorder that can be treated with gestrinone is lipedema (Figure 1), a condition that, despite being described more than 80 years ago, was recently (2022) recognized as a disease and gained notoriety in the scientific community [24]. It is still significantly underdiagnosed and is confused by many professionals who do not have much experience with the condition of obesity. Lipedema is not an exclusively female disease. However, it affects mostly women, approximately 11% of adult and post-pubescent women worldwide, as it is a pathology where there is, among the pathophysiological etiologies, a disorder in estrogen metabolism, an estrogenic predominance that manifests in periods of hormonal transition such as during puberty, postpartum, climacteric and menopause; the disorder can be hereditary [24].

Lipedema confers some physical and social limitations on affected women, as it involves edema and the abnormal distribution of fatty tissue in the limbs, causing increased pain and local sensitivity; the vasculature is affected by weaknesses, increasing the propensity for hematomas, bilateral expansion, and an asymmetrical pattern of subcutaneous adipose tissue, which are prevalent findings. We also know that in lipedema, there is an overexpression of aromatase, which increases the conversion of androgens into estrogens in adipose tissue, further fueling the pathophysiology of the disease [24]. Gestrinone could be critical in controlling this pathology by inhibiting aromatase and blocking estrogen receptors.

In our clinical experience, described in previous trials, the use of gestrinone, nestorone, and testosterone through silastic subcutaneous implants, which have controlled daily dose releases, can benefit some gynecological pathologies [20,25]. However, this therapy is carried out in a specialized center with appropriate dosages for patients with lipedema, depending on the reproductive stage of life, that improve the quality and appearance of the skin affected by the disease, in addition to improving pain and edema in affected limbs, thereby offering greater comfort and an improved quality of life.

Regarding the use of gestrinone in post menopause (Figure 1), especially in Brazil, this molecule is highly visible among progestins due to its positive aesthetic effects on patients, including muscular anabolism, improvement in the quality and texture of the skin, its action on adipocytes, and its ability to improve the metabolism of lipolysis. This has made it an attractive and desired molecule among women, especially during menopause.

One of the main hormones to be replaced during menopause is estrogen; it plays a crucial role in cardiovascular, brain, and bone health, body composition, fat deposition patterns, and skin quality, among other things. Other functions on metabolism and female health can be extended to patients over 65 [26,27].

When replacing estrogen in a woman in menopause, we are faced with the challenge of finding a balance in the dose and route of administration so that the estrogen does not cause endometrial thickening or breast proliferative action. Thus, given all the functions of gestrinone, this molecule can be a therapeutic tool with great applicability in climacteric and menopausal contexts to counterbalance the unwanted effects of estrogenic HT. Care must be taken when prescribing estrogen, especially when climacteric women have had estrogen-dependent conditions since the beginning of their reproductive life, including endometriosis, uterine fibroids, breast fibroadenomas, menstrual dysphoric disorder, and lipedema, among others. Such situations require attention and control even after the arrival of menopause. This demonstrates the therapeutic potential of gestrinone to protect against the worsening of these estrogen-dependent conditions when undergoing post-menopausal estrogen therapy.

Proof of this has been demonstrated in the most recent studies, which have shown that estrogen therapies combined with progestins confer a reduction in the risk of endometrial cancer of 45%. In comparison, the combination of estrogen and progesterone conferred a significant increase in this risk, of 33%, in a population of women 65 years or older [26].

The prevalence of endometriosis in post-menopausal women is 2–4% [28]. It is known that post-menopausal estrogenic HT may be associated with the reactivation of endometriosis; however, this is more common when the therapy is not combined with progestin [28]. Recurrence becomes more frequent in obese women, due to the conversion of androgens into estrogens by aromatase in adipose tissue, and in women with serious illnesses.

Gestrinone, as a progestin with peculiar and unique characteristics, has diverse clinical applicability post menopause. One of the biggest challenges in menopausal HT is the stability of symptoms and regularization of hormonal levels; depending on the route of administration, absorption peaks, or even an erratic form of absorption, can occur, which implies a non-resolution of the symptoms. There are three possible routes for using gestrinone (Table 2); in our clinical practice, we obtain it through oral, vaginal, and silastic hormonal implants, in continuous daily release doses that are controlled, without the occurrence of peaks, where the stable release of small doses provides excellent resolution and treatment of symptoms. It is common in our practice to combine therapy with estrogen implants, using gestrinone implants in post-menopausal patients who meet the eligibility criteria.

Regarding hormonal implant therapy, each silastic gestrinone implant consists of a thin 40 mm silicone tube containing 40 mg, and releases 110 mcg of the molecule daily [1,4]. A dose between 40 mg and 240 mg of gestrinone via silastic implants is necessary to control bleeding and to protect against endometrial thickening in these patients, as well as a therapeutic dose to control estrogen-dependent diseases [1,4,17,32,33,34,35].

However, more safety studies on gestrinone, the control of side effects, and pharmacokinetic studies on the main routes used are needed. Although controversial, the significant concerns with gestrinone are the possible harmful effects on bone mass and lipid profiles [36]. However, a recent review demonstrated the reliability and benefits of using gestrinone for a selected group of women [37], especially for those with estrogen-dependent conditions and pathologies where the hormonal implant can assist in treatment as a long-acting method (Figure 1).

Our review has some limitations, including, as follows: the scarcity of randomized clinical studies with gestrinone; the heterogeneous characteristics of patients analyzed in prospective studies; the lack of standardization of doses for the use of gestrinone; the lack of extensive recent studies; and the absence of a specific route for the use of this progestin. For this reason, it is essential to continue clinical research related to the safety of gestrinone. The current emphasis is on studying the use of subdermal implants in treating endometriosis; Brazil is the world leader in this research. The most extensive study is GLADE [38], a multicenter, prospective, randomized, double-blind, and placebo-controlled study that evaluated the safety and tolerability of subdermal implant–bioabsorbable gestrinone pellet use in women with pelvic pain secondary to endometriosis. The exploratory aim was to compare the use of a gestrinone pellet with a placebo pellet in participant satisfaction, change in pelvic pain intensity, use of rescue pain medication, quality of life, sexual function, and work activity. Another ongoing study evaluates the effects of therapy with implantable gestrinone compared to oral dienogest in relieving complaints related to endometriosis [39]. Finally, another study assessed the association of misuse of hormonal implants with gestrinone with cardiovascular health outcomes in young women [40]. Therefore, we will soon obtain more data on the doses, safety, and tolerance of gestrinone in the form of a subdermal implant. Despite this, future research is needed to assess this long-term therapy’s real risks or benefits, including the risks and benefits to post-menopausal women.

## 4. Conclusions

Gestrinone is a progestin with unique characteristics that has demonstrated its potential in the treatment of menopause and estrogen-dependent pathologies, such as endometriosis, uterine fibroids, premenstrual dysphoria, lipedema, and in climacteric patients. Due to its androgenic and antiestrogenic characteristics, it is expected that this molecule can provide benefits in improving muscle mass in patients with sarcopenia in addition to its possible antiproliferative and antiangiogenic effects, which offers an interesting perspective on the safety of this molecule and its potential therapeutic indications. We analyzed the possible application of gestrinone as a long-acting therapeutic agent through the use of hormonal implants and its benefits and potential risks. Available evidence on the safety of doses and routes is limited. Gestrinone appears to be effective compared to other progestins and may have some advantages in the treatment of estrogen-dependent pathologies. Future research must evaluate gestrinone’s long-term safety and potential therapeutic indications.

## Figures and Tables

**Figure 1 pharmaceuticals-17-01248-f001:**
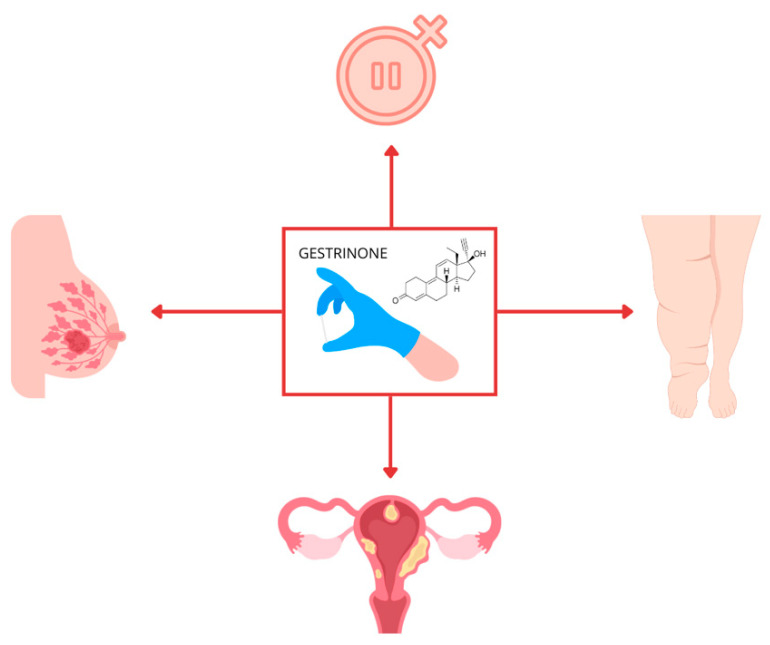
Future therapeutic possibilities of using gestrinone implants in estrogen-dependent pathologies and menopause.

**Table 1 pharmaceuticals-17-01248-t001:** Actions and equivalences of progestins to progesterone. P4: progesterone; AG: antigonadotropic; E: estrogenic; AE: antiestrogenic; A: androgenic; AA: antiandrogenic; GC: glucocorticoid; AM: antimineralocorticoid (adapted from Schindler AE and Maturitas, 2008 [2]).

Progestin/Action	P4	AG	E	AE	A	AA	GC	AM
Progesterone	+	+	−	+	−	+/−	+	+
Gestrinone	+/−	+	−	+	+	−	+	+
Nestorone	+	+	−	+/−	−	+/−	−	−
Dihydrogesterone	+	−	−	+	−	+/−	−	+/−
Medrogesterone	+	+	−	+	−	+/−	−	−
Chlormadionona Acetate	+	+	−	+	−	+	+	−
Cyproterona Acetate	+	+	−	+	−	++	+	−
Megestrol Acetate	+	+	−	+	+/−	+	+	−
Medroxyprogesterone Acetate	+	+	−	+	+/−	−	+	−
Nomegestrol Acetate	+	+	−	+	−	+/−	−	−
Promegestone	+	+	−	+	−	−	−	−
Trimegestone	+	+	−	+	−	+/−	−	+/−
Drosperinone	+	+	−	+	−	+	−	+
Nortiesteronae	+	+	+	+/−	+	−	−	−
Linestrenol	+	+	+	+	+	−	−	−
Norethisterone	+	+	+	+	+/−	−	−	−
Levonorgestrel	+	+	−	+	+	−	−	−
Norgestimata	+	+	−	+	+	−	−	−
3-Ceto-desogestrel	+	+	−	+	+	−	−	−
Gestoden	+	+	−	+	+	−	+	+
Dienogest	+	+	−	+/−	−	+	−	−

**Table 2 pharmaceuticals-17-01248-t002:** Routes, doses, and applicability of gestrinone in human trials.

Author(s), Year, Reference Number	Route	Gestrinone Protocol	Characteristic of Study/Participants	Applicability
Coutinho et al. (1975) [1]	Silastic subdermic Implant	30–40 mg (6 to 12 months)	Prospective study (531 pre-menopausal women)	Contraception effectiveness
Coutinho et al. (1984) [4]	Oral	5 mg twice a week (3 to 9 months)	Prospective study (28 women diagnosed with breast fibrocystic disease)	Fibrocystic disease of the breast
Ciou et al. (2022) [5]	-	-	Retrospective study (8330 endometriosis patients)	Anticancer efficacy of gestrinone
Gestrinone Italian Study Group (1996) [15]	Oral	2.5 mg twice a week versus leuprolide acetate 3.75 mg depot injections every 4 weeks	Randomized, double-blind, multicenter study (25 endometriosis patients with moderate or severe pelvic pain)	Pelvic pain associated with endometriosis
Coutinho et al. (1989) [18]	Oral/Vaginal	Oral 2.5 mg of gestrinone three times weekly Oral 5.0 mg twice weekly Vaginal route tablets containing 5 mg (24 months)	Randomized, prospective study (100 women with leiomyomas)	Reduction in uterine volume
Alvarez et al. (1978) [19]	Silastic subdermic Implant	Gestrinone 30 mg versus levonorgestrel 30 mg	Prospective study (100 pre-menopausal women)	Contraception effectiveness
Dawood et al. (1997) [23]	Oral	1.25 mg or 2.5 mg twice a week (24 weeks)	Randomized, double-blind, prospective study (11 patients given gestrinone	Treatment of endometriosis
Cunningham et al. (1987) [29]	Oral	2.5 mg every 3 days	Prospective study (15 patients with locally advanced or metastatic breast cancer)	Anticancer efficacy of gestrinone
Hornstein et al. (1990) [30]	Oral	1.25 mg twice weekly 2.5 mg twice weekly	Randomized, double-blind, prospective study (12 women with endometriosis	Treatment of endometriosis
Venturini et al. (1989) [31]	Oral	2.5 mg twice weekly (6 months)	Prospective study (11 women with mild or moderate endometriosis)	Treatment of endometriosis
Coutinho et al. (1988) [32]	Vaginal	2.5 mg tablets weekly 2.5 mg twice a week three 2.5 mg tablets weekly (6 to 8 months)	Prospective study (110 patients with endometriosis).	Treatment of endometriosis
Coutinho et al. (1989) [33]	Oral/Vaginal	2.5–5 mg (orally or by vaginal pessary), two or three times weekly (6 months)	Prospective study (300 women with uterine myomas)	Treatment of uterine myomas
Coutinho et al. (1986) [34]	Oral/Vaginal	Oral 5 mg twice weekly Oral 2.5 mg capsules three times weekly Vaginal 2.5 mg tablets three times weekly (4 to 13 months)	Prospective study (97 women, with uterine leiomyomas)	Treatment of uterine leiomyomas
Zhang et al. (2016) [35]	Oral	Oral 2.5 mg twice a week versus mifepristone 12.5 mg daily	Randomized controlled prospective study (65 women with endometriosis)	Treatment of endometriosis
Fukuda et al. (1989) [36]	Oral	5 or 10 mg weekly (4 to 6 months)	Prospective study (12 women with endometriosis)	Effect on serum lipid and lipoprotein levels in women with endometriosis
De Souza Pinto et al. (2023) [37]	-	-	Systematic review and meta-analysis of 16 studies involving 1286 women	Treatment of endometriosis

## Data Availability

The datasets used and/or analyzed during the current study are available from the corresponding author upon reasonable request.

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
