# Peer review of "Effects, Doses, and Applicability of Gestrinone in Estrogen-Dependent Conditions and Post-Menopausal Women"

_pharmaceuticals, 2024, doi:10.3390/ph17091248_

Round 1
Reviewer 1 Report
Comments and Suggestions for Authors
The article “Effects, doses, and applicability of Gestrinone in estrogen-de-pendent conditions and postmenopausal women” is a prospective type article but according to instruction guidelines (https://www.mdpi.com/journal/pharmaceuticals/instructions) we have only the original research article and review type. The current article consists to parts introduction and conclusion parts. in regular each type consists Introduction, a Result, a Discussion, Materials and Methods, and a Conclusion.
The article:
1- The language needs minor corrections for grammar errors. some s, have/has, this these error detected
2- what are your include and exclude parameters
3- how much data/article/source were evaluated.
4- line 83 MTT is not a cell metabolic activity test please check.
5- Gestrinone mechanism of action and effect on anticancer mechanism needs more descriptive sentences.
Comments on the Quality of English Language
The language needs minor corrections for grammar errors. some s, have/has, this these errors were detected. please check.
Author Response
Dear Reviwer,
Q0: The article “Effects, doses, and applicability of Gestrinone in estrogen-de-pendent conditions and postmenopausal women” is a prospective type article but according to instruction guidelines (https://www.mdpi.com/journal/pharmaceuticals/instructions) we have only the original research article and review type. The current article consists to parts introduction and conclusion parts. in regular each type consists Introduction, a Result, a Discussion, Materials and Methods, and a Conclusion.
A0: We agree with the reviewer, and we thank you for the opportunity to clarify it. We work exhaustively on improving the manuscript's limitations, as follows.
Q1: The language needs minor corrections for grammar errors. some s, have/has, this these error detected
A1: We agree with the reviewer, and we thank you for the opportunity to clarify it. We carry out a general review of grammar errors and corrections.
Q2: what are your include and exclude parameters
A2: We agree with the reviewer, and we thank you for the opportunity to clarify it. We included all studies related to Gestrinone according to the criteria included in section “2. Materials and methods” between pages 87 to 98.
Q3: how much data/article/source were evaluated.
A3: We agree with the reviewer, and we thank you for the opportunity to clarify it. Our review is not a systematic but a narrative review. Narrative literature review articles are publications that describe and discuss the state of science on a specific topic or theme from a theoretical and contextual point of view. These review articles, like ours, do not list the types of databases and methodological approaches used to conduct the review, nor the evaluation criteria for including articles retrieved during the database search. Even so, in response to the reviewer's request, we included it in section “2. Materials and methods” between pages 87 to 98.
Q4- line 83 MTT is not a cell metabolic activity test please check.
A4: We agree with the reviewer, and we thank you for the opportunity to clarify it. We adjusted the text between pages 110 and 119.
Q5: Gestrinone mechanism of action and effect on anticancer mechanism needs more descriptive sentences.
A5: We agree with the reviewer, and we thank you for the opportunity to clarify it. We adjusted the text and included a more descriptive anticancer mechanism of gestrinone between pages 120 and 155.
Q6: The language needs minor corrections for grammar errors. some s, have/has, this these errors were detected. please check.
A6: We agree with the reviewer, and we thank you for the opportunity to clarify it. We carry out a general review of grammar errors and corrections.
Thank you for considering this manuscript.
Sincerely,
Guilherme Renke
Reviewer 2 Report
Comments and Suggestions for Authors
The manuscript presents an informative review of Gestrinone (R-2323), detailing its therapeutic uses, dosing strategies, and potential applications in estrogen-dependent conditions, especially in postmenopausal women. The authors have successfully synthesized existing literature, providing a comprehensive overview of the subject.
Abstract: The abstract succinctly summarizes the key points. However, it could be enhanced by explicitly stating the main conclusions drawn from the review.
Figures and Tables: The inclusion of tables summarizing dosing regimens is beneficial. Improving the labeling and referencing of figures in the text would enhance clarity.
Conclusion: The conclusion effectively summarizes the potential benefits of Gestrinone but should also stress the need for further research to address identified gaps.
Suggested Revisions:
1- Critically evaluate the methodologies of cited studies to assess their reliability.
2- Expand the discussion on side effects and long-term safety concerns.
3- Update the reference list to include more recent research findings.
4- Revise the abstract and conclusion to provide clearer emphasis on future research directions.
Author Response
Dear Reviewer,
Q0: The manuscript presents an informative review of Gestrinone (R-2323), detailing its therapeutic uses, dosing strategies, and potential applications in estrogen-dependent conditions, especially in postmenopausal women. The authors have successfully synthesized existing literature, providing a comprehensive overview of the subject.
A0: We agree with the reviewer, and we thank you for the opportunity to clarify it. We work exhaustively on improving the manuscript's limitations, as follows.
Q1: Abstract: The abstract succinctly summarizes the key points. However, it could be enhanced by explicitly stating the main conclusions drawn from the review.
A1: We agree with the reviewer, and we thank you for the opportunity to clarify it. We carry out a general review on the abstract between pages 9 and 19.
Q2: Figures and Tables: The inclusion of tables summarizing dosing regimens is beneficial. Improving the labeling and referencing of figures in the text would enhance clarity.
A2: We agree with the reviewer, and we thank you for the opportunity to clarify it. We detailed all included studies on table 2 related to Gestrinone and we referenced the figures in the text enhancing clarity.
Q3: Conclusion: The conclusion effectively summarizes the potential benefits of Gestrinone but should also stress the need for further research to address identified gaps.
A3: We agree with the reviewer, and we thank you for the opportunity to clarify it. As requested, we carry out a general review on the conclusion between pages 430 and 442.
Suggested Revisions:
Q4: Critically evaluate the methodologies of cited studies to assess their reliability.
A4: We agree with the reviewer, and we thank you for the opportunity to clarify it. We included all studies related to Gestrinone according to the criteria included in section “2. Materials and methods” between pages 87 to 98.
Q5: Expand the discussion on side effects and long-term safety concerns.
A5: We agree with the reviewer, and we thank you for the opportunity to clarify it. We included the discussion on side effects and long-term safety concerns related to Gestrinone according to the text between pages 360 and 422.
Q6: Update the reference list to include more recent research findings.
A6: We agree with the reviewer, and we thank you for the opportunity to clarify it. We included more recent trials in references 8-13; 17; 35; 38-40.
Q7: Revise the abstract and conclusion to provide clearer emphasis on future research directions.
A7: We agree with the reviewer, and we thank you for the opportunity to clarify it. As requested, we carry out a general review providing a clearer emphasis on future research directions on the abstract between pages 9 and 19 and conclusion between pages 430 and 442.
Round 2
Reviewer 1 Report
Comments and Suggestions for Authors
The revised answer is adequate. Thank you
Reviewer 2 Report
Comments and Suggestions for Authors
Accepted